# Role of Neuroglia in the Habenular Connection Hub of the Dorsal Diencephalic Conduction System

Anton J. M. Loonen 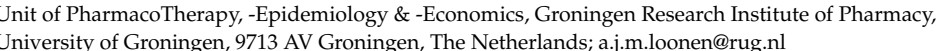

Unit of PharmacoTherapy, -Epidemiology & -Economics, Groningen Research Institute of Pharmacy, University of Groningen, 9713 AV Groningen, The Netherlands; a.j.m.loonen@rug.nl

**Abstract:** Astrocytes and microglia play important roles in organizing the structure and function of neuronal networks in the central nervous system (CNS). The dorsal diencephalic connection system (DDCS) is a phylogenetically ancient regulatory system by which the forebrain influences the activity of cholinergic and ascending monoaminergic pathways in the midbrain. The DDCS is probably important in inducing aspects of mental disorders, such as depression and addiction. The habenula is the small but highly complex connecting center of the DDCS in the epithalamus that consists of a medial (MHb) and lateral (LHb) division. MHb and LHb are built differently and connect different brain structures. Studies in animal models and human biomarker research provide good evidence that astroglia and microglia also affect the symptoms of mental disorders (such as depression). The significance of these neuroglia in habenular neurotransmission has not been extensively studied. This review article provides arguments for doing so more thoroughly.

**Keywords:** astrocyte; microglia; habenula; mood disorders; anxiety disorders; addiction; schizophrenia; metabolic syndrome

## 1. Introduction

Although the connections of the habenula were first described at least as early as 1925 [1] and this nuclear complex has even been found in the ancestors of vertebrates [2], it is only in the last decade that serious attention has been given to the important role of the dorsal diencephalic conduction system (DDCS) in the development of various mental diseases [3]. This means that the DDCS can be considered one of the oldest descending control systems by which the forebrain is linked to the midbrain. In the midbrain, we find the main ascending monoaminergic pathway systems that, among other things, determine the output of the forebrain. The DDCS is, therefore, an important means through which the forebrain regulates its own activity. Since human behavior can be considered either one of the products of the forebrain or dependent upon the essential activity of the forebrain, it is actually very obvious that the DDCS also plays a role in mental illnesses in which the intensity of behavior is altered.

The research by Bartrop et al. [4] on the influence of grief on the function of B and T cells and hormone levels in the spouses of deceased patients initiated what would later develop into psychoneuroimmunology [5–7]. Currently, the connection between the functioning of the immune system and the development of mental disorders is widely accepted [8–11]. This is especially well conceived for depression [12–15], schizophrenia [16–18] and post-traumatic stress disorder [19–21]. The red line here is that aspects of mental disorders are associated with neuroinflammatory processes that, in turn, tend to be attributed to the action of cytokines.

Cytokines are usually relatively small proteins of ~5–25 kDa with important autocrine, paracrine or humoral regulatory functions within the immune system [13,14,22]. They bind to specific, high affinity, cell surface receptors and have pleiotropic effects, which means that they have multiple actions and affect multiple target cells (other immune

cells, but in the central nervous system (CNS) also neuroglia and neurons). Cytokines are produced by a variety of immune cells, as well as neurons within the CNS. Peripherally produced cytokines can also signal to the brain, for instance, by crossing the blood–brain barrier through saturable transporters, by humoral transmission via the circumventricular organs, and by neural transmission via afferent nerves (e.g., the vagal nerve). CNS cytokine receptors are predominantly activated by cytokines that are produced locally in the brain in response to peripheral cytokines. In addition, they interact with the endocrine system (e.g., the adrenocorticoid regulation) and affect neuronal processes indirectly. Cytokines are also well known to be involved in brain development and neuronal plasticity [13,14,22].

In this review article, I want to focus mainly on the possible role of neuroglia in regulating the activity of the DDCS in the habenula. This will build on the anatomical and functional information of a recently published article on developing appropriate biomarkers for the function of the DDCS [23].

## 2. Theoretical Background for a Role of the DDCS

Over the last six years, Svetlana Ivanova and I have incrementally developed a new theory of the biological mechanisms underlying mental disorders such as depression, addiction and aspects of psychosis [3,23–25].

We took as our starting point the evolutionary development of the vertebrate's brain and on this basis, distinguished between a primary, secondary and tertiary segment of the forebrain [23,25]. As indicated in Table 1, the human primary forebrain consists mainly of the amygdaloid complex, hippocampal complex, posterior septum, hypothalamus and epithalamus, and initiates the emotional response (appetitive, aggressive, defensive, sexual, etcetera). In humans, the secondary segment consists of some limbic parts of the frontal cortex, ventral extrapyramidal system (limbic cortex, nucleus accumbens, ventral pallidum, part of the dorsal thalamus) and epithalamus, and regulates the readiness and intensity of this emotional response. The tertiary part consists of the rest of the forebrain (isocortex, nucleus caudatus, putamen, globus pallidus, dorsal thalamus) and processes sensory information, generates the most adequate output, and is necessary for thoughts, fantasies, memories and awareness [23,25].

**Table 1.** Role of the DDCS within the context of forebrain functioning.

|  | Anatomical Components | Circuits | Role |
|---|---|---|---|
| Primary forebrain * | Corticoid amygdala<br>Extended amygdala<br>Hippocampal complex<br>Posterior septum<br>Hypothalamus<br>Habenuloid complex | DDCS | Initiates the emotional response |
| Secondary forebrain | Limbic cortex<br>Nucleus accumbens<br>Ventral pallidum<br>Thalamus<br>Hypothalamus<br>Habenuloid complex | Ventral CSTC<br>(aCg re-entry)<br>(sCg re-entry)<br><br>DDCS | Regulates intensity<br>of the emotional response |
| Tertiary forebrain | Isocortex<br>Putamen<br>Caudate nucleus<br>Globus pallidus<br>Dorsal thalamus | Cortical<br><br>Dorsal CSTC<br><br>Intracerebral<br>Cerebrospinal | Initiates and regulates of the<br>voluntary and planned response |

* Tertiary forebrain includes secondary forebrain, which includes primary forebrain, as in a Russian Doll model. The ventral CSTC circuits contain aCg re-entry circuits regulating the intensity of reward-seeking behavior (regulating 'Pleasure') and sCg re-entry circuits regulating the intensity of adversity-avoiding behavior (regulating 'Happiness'). aCg is anterior cingulate gyrus; CSTC is Cortical-Striatal-Thalamic-Cortical; DDCS is Dorsal Diencephalic Connection System; sCg is subgenual cingulate gyrus.

The activity of the basal ganglia (nuclear amygdala, nucleus accumbens, putamen, caudate nucleus) and cerebral cortex (hippocampal complex, corticoid amygdala, neocortex) are regulated in part by ascending monoaminergic (dopamine, serotonin, histamine, norepinephrine) pathways from the midbrain and caudal hypothalamus [26–28]. These, in turn, are partly controlled by the DDCS, with the habenula in the epithalamus as its main connecting hub [24,29]. The ventral extrapyramidal circuits are important structures that are under the control of the ascending dopaminergic and serotonergic neuronal pathway systems [14]. These parallel ventral cortico-striatal-thalamic-cortical circuits include a series of re-entry circuits that begin and end in a continuum of limbic cortical areas from the anterior cingulate cortex (aCg) to the subgenual cingulate cortex (sCg), and possibly through to the anterior part of the insular cortex (aIns) [23]. Two sets of these parallel re-entry circuits regulate the intensity of reward-seeking and adversity-avoiding behaviors, respectively [23,30]. When reward-seeking behavior is successful, it produces feelings of pleasure, and in the case of adversity-fleeing behavior, it produces feelings of happiness. We have, therefore, named these circuits as "circuits regulating pleasure and happiness" [30]. The regulation of these behaviors plays an important role in depression, anxiety and addiction, and along these lines, the DDCS and, therein, the habenula, probably plays an important role in their emergence [3,24].

## 3. Anatomy of the DDCS

For a more detailed description of the anatomical construction and connections of the habenula, reference can be made here to several recent review articles [29–36]. I will suffice here with an overall sketch, as shown in Figure 1.

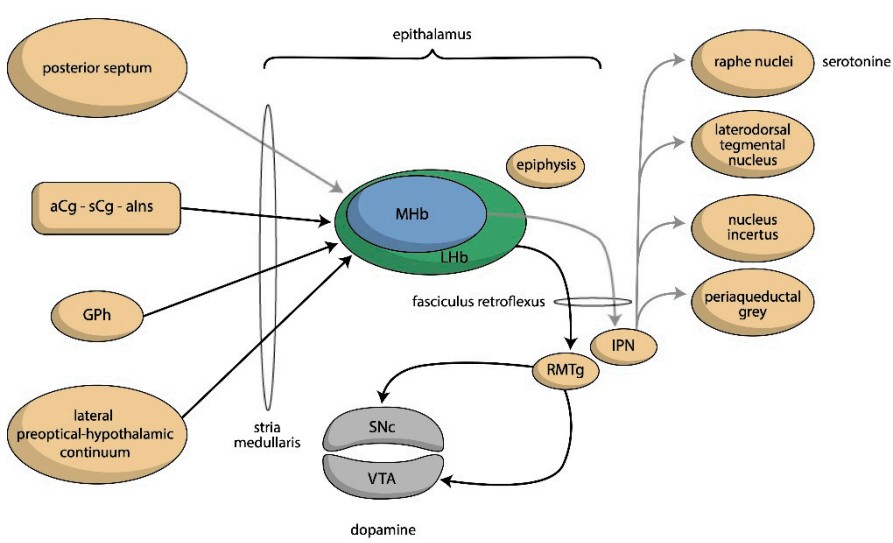

**Figure 1.** Anatomy of the central part of the dorsal diencephalic connection system (DDCS). GPh—human equivalent of the habenula-projecting pallidum; IPN—interpeduncular nucleus; LHb—lateral habenula; MHb—medial habenula; RMTg—rostromedial tegmental nucleus; SNc—Substantia nigra, pars compacta; VTA—ventral tegmental area.

The central part of the DDCS consists of the stria medullaris, habenula and fasciculus retroflexus. The *stria medularis* can be considered the input channel of the DDCS and the habenuloid complex is the center in which the input is switched onto the neurons of the output channel: the fasciculus retroflexus. The stria medullaris, the habenuloid complex, the connections between the left and right DDCS (i.e., the habenular commissure) and the endocrine epiphysis (pineal gland) form the so-called **epithalamus** (see Figure 1). This is a phylogenetically ancient part of the anatomical thalamus (diencephalon) on the

dorsal and posterior side of the dorsal thalamus, which clinicians commonly refer to as "*the*" thalamus.

The *fasciculus retroflexus* gets its name from a sharp backward bend in its course around/in the red nucleus to the unpaired interpeduncular nucleus in the tegmentum of the midbrain [35]. Connections through this fasciculus retroflexus have changed little to not at all during the course of evolution [29].

The *habenula*, or rather the habenuloid complex, consists of a small medial division and a much larger lateral one (~5% vs. 95%). It is possible that the medial part is phylogenetically somewhat older, being already prominent in a representative of a more primitive ancestor of vertebrates: the hagfish [2]. In rodents, the compact medial division comprises 5 subnuclei and the looser lateral one comprises at least 10 [34]. In many animal species, the habenula is asymmetrically shaped. While this is less the case in humans, there is evidence of lateralization in function in humans [37,38].

As shown in Figure 1, the **Medial Habenula (MHb)** receives most of its input from the posterior septal nuclei, i.e., the triangular nucleus and bed nucleus of the anterior commissure, and mainly provides output to the *interpeduncular nucleus* (IPN). This IPN is an unpaired, integrating center in the bottom of the upper midbrain and is richly connected via ascending and descending projections to various forebrain and other brainstem structures [39,40]. It plays a key role not only in modulating behavior and cognition, but through connections with the hippocampus also in the formation of memory [41,42]. Where the input from the posterior septal nuclei comes from has not yet been extensively investigated. From ontogenetic studies, connections with the MHb on the one hand and the hippocampus on the other can be deduced [43,44]. In any case, the posterior septum is not part of Gray and McNaughton's Septo-Hippocampal Regulation System (SHS) [45] which, as a "Behavioural Inhibition System," plays a role in the regulation of anxiety and especially fear [46].

The **Lateral Habenula (LHb)** receives most of its input from a continuum of nuclei that extends from the *lateral preoptic area to the lateral hypothalamic region* and likely also includes the ventral pallidum and bed nucleus of the stria terminalis [31]. Output from the lateral habenula travels mainly to the inhibitory *rostromedial tegmental nucleus* (RMTg) of the midbrain, in addition to several monoaminergic centers, the laterodorsal tegmental nucleus and the incertus nucleus. The output of the RMTg inhibits the activity of the ascending dopaminergic pathways to the striatum and cerebral cortex, which originate in the *substanta nigra, pars compacta* (SNc) and *ventral tegmental area* (VTA). Moderate input from the LHb originates from a continuum of limbic prefrontal areas extending from the *anterior cingulate cortex* (aCg) through the *subgenual cingulate cortex* (sCg) to the *anterior insular cortex* (aIns). This aCg–sCg–aIns continuum also provides input to the accumbens nucleus and thus the ventral CSTC circuits described in Section 2. The entopeduncular nucleus (*globus pallidus* interna in primates) also provides input to the LHb and this probably corresponds to the input from the habenula-projecting pallidum in the most primitive vertebrates (*GPh*) [47]. These last two structures in particular, I believe, can be considered as being greatly significant in the development of mood and anxiety disorders, as well as addiction. Moreover, many fibers give feedback to the LHb from various brainstem nuclei [31,34].

The connections from and to the habenula are certainly not limited to the connectivity described above. This is evidenced in part by the recent review by Roman and colleagues [35], who systematically searched the scientific literature for relevant publications. However, the value of their results have characteristic limitations. The number of investigations of the situation in humans is very limited and there are important species differences, for example, in relation to lateralization. Moreover, the studies are sometimes relatively old and the techniques used were of a different order to those of newer studies that use genetically modified laboratory animals. Finally, it is often difficult to determine what value should be assigned to the connections described between them. Put somewhat bluntly, in the brain, virtually every structure is interconnected back and forth with every

other structure. How, then, does one determine whether the connections described are also clinically relevant? Of course, these comments are only meant to outline the scope of the problem. Therefore, there is still more than enough to explore (again). In particular, the origins of the input of the MHb and the role it plays in the development of the symptoms of mental illnesses have not yet been properly explored at all.

## 4. Putative Role of Neuroglia in the Habenula

### *4.1. Astroglia*

That neuroglia may be of essential significance for the behavioral regulatory function of the DDCS has received relatively little attention to date. The attention given to it has focused mainly on the role of astroglia. It is known that in the MHb, a different type of astrocyte is found to that in the LHb [33,48]. In the MHb, protrusions of polar astrocytes form a tight sheath around neuronal cell bodies and bundles of axons. In the LHb, mainly stellate astrocytes are found, which are also dominant in other forebrain regions. The non-stellar MHb astrocytes express an ectonucleotidase that is relatively specific for ATP. This is consistent with the release of ATP as a fast neurotransmitter from afferent fibers from the posterior septum [49]. The star-shaped astrocytes of the LHb have a multitude of extensions to all sides (Figure 2). Very well-known are the ramifications to (glutamatergic) synapses, which are found in various places in the forebrain. They are also known to me, for example, for their significance in inducing neuroplastic changes in so-called "striatal spine modules" in the extrapyramidal system [50,51]. Part of these so-called tripartite synapses are perisynaptic astrocytic processes (PAPs), which are very thin ends (frequently below 100 nm thick, compared to the optimal width of the synaptic cleft of 12–20 nm [52]) of the elaborate ramifications of these extensions [53–55]. These ramifications of stellate astrocytes are so extensive that in total, these PAPs occupy about 80% of the astrocytic plasma membrane [53] and in a mouse brain, a single astrocyte is estimated to contact more than 100,000 synapses; meanwhile, in a human brain, this may be the case but with more than a million synapses [55]. In addition, this fine structure is very dynamic [54].

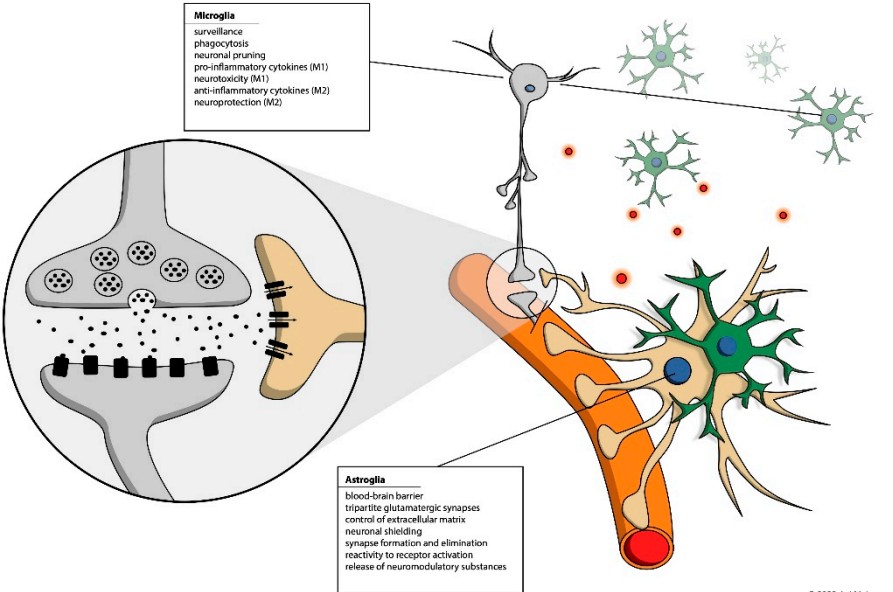

**Figure 2.** Schematic representation of the relationship between microglia, astroglia and neurons in the lateral habenula (LHb). The stated functions of microglia and stellate astrocytes are not meant to be exhaustive. The magnification shows the structure of the tripartite synapse schematically. In these tripartite synapses, perisynaptic astrocytic processes (PAPs) modify neurotransmission by removing neurotransmission from the synaptic cleft. This process is modified by certain cytokines. The figure is an adaptation of the figures of Aizawa et al. [33] and Chen et al. [56].

Much has now been published on how astrocytes can influence glutamatergic neurotransmission (for reviews see [55,57–60]). Incidentally, most research does not refer to tripartite synapses in the lateral habenula [59], but it is likely that the processes are also at play there; the DDCS is rich in glutamatergic fibers [23]. In some of these, glutamate is present as a co-transmitter. Three processes are mentioned, through which PAPs may influence excitatory transmission in glutamatergic synapses [59]: "morphological mechanisms of astroglial plasticity," "expression of astroglial membrane transporters and receptors," and "gliotransmission." Incidentally, GABAergic neurotransmission can also be targeted [59]. I have previously called attention to the possibility that a shift from glutamatergic to GABAergic transmission or its reverse within, for example, the pallido-habenular GPh fibers, may dramatically affect the intensity of reward-seeking and misery-avoiding behavior [23].

Something that is fairly odd is that so little attention has been paid to the possible role of cytokines in causing astroglial-induced changes in the function of the DDCS. After all, astrocytes are known to be of great significance in all sorts of processes that cause severe neuroinflammation [61–65], and so why would this not play a role in the lighter forms that are obvious in mental disorders [7,10,21]? Although microglia may be considered the dominant immunocompetent cells of the CNS, astrocytes participate fully in inducing and mediating the neuroinflammatory responses. Upon activation by pro-inflammatory stimuli, astrocytes produce a variety of cytokines such as interleukin-6 (IL-6), interleukin-10 (IL-10), interleukin-17 (IL-17), interleukin-1β (IL-1β), and tumor necrosis factor (TNF) [58]. These can have direct effects on the function and integrity of glutamatergic synapses [58]. Moreover, shape changes might occur, which likely undermine the support function of PAPs. The question remains, of course, as to what extent these kinds of neurotoxic changes also occur in mental illnesses such as depression. This is certainly not ruled out, for example, given the increased incidence of Alzheimer's disease in sufferers of depression [66]. Confounding is the dominant occurrence of interleukin-18 (IL-18) in the neurons of the superior nucleus of the MHb complex [67,68]. IL-18 plays an important role in the immune system [69], but neuronal IL-18 of the MHb and other structures may also have a distinct role [70].

*4.2. Microglia*

Microglial cells are the major immune cells in the CNS but apart from their role in neuroinflammation, they have many other functions in the healthy and diseased brain [71–74]. They are macrophages embryologically descended from mesenchymal tissue in the yolk sac [75]. In the adult brain, they form a pool of macrophages sealed from the blood that is stable in size but self-renewing at a specific turnover rate. Due to changes thereof, the pool may exhibit regional and temporal variations in size and composition [76]. The foregoing also means that the composition and role of microglia in influencing the habenula and thus the DDCS should be considered not as a separately fixed, but as a dynamic fact. Microglia have an important function during embryonic development. Not only do they clear away all kinds of debris as macrophages, but they also play an important role in controlling the number of neurons and the existence of neuronal connections by removing unused synapses and other parts of redundant neuronal connections, a process called "synaptic pruning" [74]. They have a function in the adult brain as macrophages, but they are also involved in protecting the integrity of the CNS through a function called "surveillance" [77,78]. In this process, homeostatic microglia constantly scan a relatively wide area with their highly motile and ramified expansions for invading microorganisms, malfolded proteins such as amyloid β, chemokines and cytokines, metabolites, inorganics, as well as changes in the pH of the extracellular matrix, etcetera [72,79]. Upon relevant activation, microglia can remove damaged cells, as well as dysfunctional synapses, in a process termed "synaptic stripping" [80]. Hence, microglia constantly interact with neurons, oligodendrocytes and astrocytes, and thus play an important role in accomplishing neuroplastic changes [81]. In addition, upon corresponding activation, microglia have an important role as immunocompetent cells [82–86]. Depending on the type of activation, the same microglia

cell can make the transition into a pro-inflammatory (M1) or anti-inflammatory (M2) state, and after prolonged activation, into a regressive declined state. The first state is induced by pro-inflammatory signals (such as Toll-like receptor (TLR) agonists related to infection, or pro-inflammatory cytokines such as tumor necrosis factor-α (TNF-α), interleukin-1β (IL-1β) or interferon-γ (IF-γ)) and produce pro-inflammatory cytokines such as TNF-α and IL-1β themselves, interleukin-6 (IL-6), nitric oxide (NO) and reactive oxygen species (ROS). The second state is induced by interleukin-4 (IL-4) and interleukin-13 (IL-13), and is associated with the production of anti-inflammatory cytokines such as IL-4 and IL-13 themselves, as well as interleukin-10 (IL-10) and transforming growth factor β (TGF-β). M2 microglia exert a variety of additional neuroprotective effects, such as promoting the degradation of misfolded and aggregated proteins, promoting neurite growth, enhancing neurogenesis, and enhancing phagocytic activity [85]. The overactivation of microglia is believed to result in the third state, which is accompanied by dystrophy and a susceptibility to death [86].

The scientific literature on the possible involvement of microglia in the pathogenesis of mental diseases has taken off significantly and has been summarized in several recent review articles [8,9,74,86–93]. When I limit myself to the significance of microglia in the development of major depression, the results of a large group of studies in both humans and experimental animals point to the critical role of microglia (for review, see [74,86]). Some of these processes take place in the LHB; illustrative is the significance of the effects of the rapidly acting antidepressant ketamine that is believed to be accomplished by affecting the functioning of this lateral complex [23]. Ketamine is a racemic mixture and is listed as a noncompetitive N-Methyl-D-Aspartate receptor (NMDA) antagonist. The R-form (arketamine) has less affinity for the NMDA receptor, but does have better and more long-lasting antidepressant effects according to animal studies and a small open trial [94–96]. This and other findings make it increasingly unlikely that the antidepressant effect of ketamine is due in any essential way to blocking NMDA receptors [97]. What it is, then, is, to date, not so clear. It has been suggested that the activation of the expression of transforming growth factor β1 (TGF-β1) by microglia and the subsequent release of brain-derived neurotrophic factor (BDNF) underlie the antidepressant action of arketamine [97]. Other evidence also exists for the involvement of microglia in mediating the influence of the lateral habenula on depression. These are mostly animal experimental studies involving, for example, chronic unpredictable mild stress (CUMS) [98] or repeated social defeat stress (R-SDS) [99] models for inducing depression. These models probably by no means cover the full pathology of major depression. Nonetheless, these models are suitable for examining immune system involvement. The study by Guan and others [99] provides a nice clue to the essential involvement of microglia activation in bringing about the response of neurons in the LHB. Within the functioning of the LHb a link between depression and addiction can also be observed [3]. An illustration of this is found in the effects of abstinence from morphine [100]. The neuronal effects in the LHb require TNF-α release and neuronal TNF receptor 1 activation, and changes also occur in LHb microglia.

Overlooking these findings, it can be concluded that while there is some evidence for the involvement of LHb microglia function in the pathogenesis of symptoms of depression, it is also true that this has not actually been investigated in sufficient detail.

## 5. Cytokines, Habenula and Psychiatric Diseases

### 5.1. Limitations of Peripheral Cytokine Levels

In human neuroscience research, links between the serum levels of cytokines and the presence and severity of mental disorders are often sought. In doing so, it is actually not entirely clear what meaning can be attached to the measured levels. It may be a peripheral immune response that is only indirectly or not at all related to the mental disorder. For example, we were thrown for a loop by immune responses due to alcohol abuse (e.g., in the digestive tract due to liver damage or altered intestinal permeability) in a comparative study of cytokine levels in depression in individuals with and without acute alcohol

dependence [101]. Moreover, in the periphery, the immune, endocrine and autonomic nervous systems are closely intertwined, and the cerebral and peripheral immune systems are by no means independent of each other [14,22,102]. Nevertheless, and partly because of this, I am relatively positive about the likelihood that the serum levels of cytokines do say something about what is happening in the CNS. There are also several possibilities through which neuroimmune signals can reach brain cells isolated from the periphery by the blood–brain barrier (BBB) [102–104]. In addition to directly crossing the BBB, e.g., via specific saturable transporters [103] and the passing of intact activated leukocytes from the blood into the brain tissue [102,105], attention is also drawn to the role of sensory signals via the vagal nerve, initiated by the peripheral binding of certain pro-inflammatory cytokines (TNF-$\alpha$ and IL-1$\beta$) [102,104]. This also implies that the same neuroinflammatory processes are probably taking place peripherally and cerebrally at least tendonally.

*5.2. Peripheral Cytokine Levels in Depression and Addiction*

The involvement of the LHb as an "anti-reward center" in the mechanism of depression and addiction is especially well elaborated [106–108]. Of note, as part of the addiction process, negative affect (depressive mood) is an obligatory part or phase [109,110]. The serum levels of cytokines in people with major depression have also been measured in several studies. In the meta-analysis by Dowlati et al. [111] of results from a total of 24 studies in people with major depression (MD) compared to a healthy control group (HC), significantly higher TNF-$\alpha$ (MD/HC = 438/350 patients, $p < 0.00001$) and IL-6 (MD/HC = 492/400, $p < 0.00001$) were found [101]. These findings were reproduced in the more recent meta-analysis by Köhler et al. [112] of a total of 82 studies, which incidentally does point out the large heterogeneity of the studies on TNF-$\alpha$ (MD/HC = 1620/1457, $p < 0.001$) and IL-6 (MD/HD = 1587/1183, $p < 0.001$). Strikingly, the differences were not significant for IL-1$\beta$ (MD/HC = 779/727, $p = 0.847$) and a significant negative association existed with IFN-$\gamma$ (MD/HC = 700/770, $p = 0.043$). There exists a meta-analysis of the results of 16 studies on peripheral levels of cytokines in alcohol use disorders (AUD) [113]. This describes that the stage of alcohol abuse and the amount of alcohol has an important influence on the measured levels. Although in this meta-analysis the levels of all measured cytokines are average and those of, for example, TNF-$\alpha$, IFN-$\gamma$ and IL-6 are specifically higher in AUD patients than in healthy controls, I do not dare to draw so many conclusions from this in connection with the many peripheral effects of alcohol itself [101]. That problem is perhaps somewhat less of an issue with disorders in terms of the use of other compounds. In the systematic review and meta-analysis by Wei et al. [114] of a total of 74 studies, significant differences were found for TNF-$\alpha$ (substance use disorder (SUD)/HC = 1324/1061, $p = 0.011$) and IL-6 (SUD/HC = 1617/1514, $p = 0.000$), but again not for IL-1$\beta$ (SUD/HC = 325/284, $p = 0.297$) and IFN-$\gamma$ (SUD/HC = 509/410, $p = 0.758$). Because the between-study heterogeneity was significant and this meta-analysis simply included AUD studies, this meta-analysis should be redone for only people who either abuse cocaine or opioids, but these patients are not easy to find. The heterogeneity of the TNF-$\alpha$ data did decrease significantly in the meta-analysis by Wei et al. when only alcohol- or cocaine-abusing patients were considered, but remained high in the case of the opioid-abusing subgroup [114]. In both meta-analyses in people with major depression and those abusing alcohol or other substances, tobacco smoking should be taken into account as a major confounder. Tobacco smoking has important pro-inflammatory effects for many reasons [115], and often smokers, co-smokers and non-smokers are indistinguishable in this type of study.

*5.3. Peripheral Cytokine Levels and Metabolic Syndrome*

A slight oddity here is to call attention to the involvement of neuroinflammatory effects in metabolic syndrome. Whether neuroinflammatory processes are also involved in schizophrenia has been extensively studied and, based on the results, is also very likely [116–118]. Based on measurements in treatment-naive people with schizophrenia, it

can be concluded that a variety of changes in peripheral cytokine levels are involved in the condition [119–121]. The same is true in first-episode psychosis with respect to white blood cell counts [122]. However, it is much more difficult to distinguish the effects on the DDCS from those in other regions of the forebrain. This is different in metabolic syndrome, as I will explain later.

Metabolic syndrome is characterized by central obesity, hyperglycemia, dyslipidemia and hypertension [123], is common in people with schizophrenia [124–126] and represents a serious burden that may be associated with their premature death [125,127,128]. Studies in rats have shown that the second-generation antipsychotic olanzapine, in addition to increasing body weight, does increase TNF-α, IL-6, and IL-1β in the plasma and white adipose tissues of rats, but not in the tissue of the frontal cerebral cortex [129]. Boiko et al. [130] found that people with an acute exacerbation of schizophrenia had a significant reduction in their (possibly initially elevated) levels of TNF-α, IL-6, IL-1β and IFN-γ; this was only in those without metabolic syndrome when treated for 6 weeks with second-generation antipsychotics (none of them received clozapine). A meta-analysis of 13 studies in 10–19-year-old adolescents found that peripheral levels of leptin, CRP, IL-6, IL-10, IL-18, fibrinogen, TNF-α, and adiponectin correlated with the components of MetS [131]. Mednova et al. studied 110 patients with schizophrenia, 46 of whom (41.8%) had MetS; this study obtained similar results for leptin, but not for adiponectin, TNF-α and IL-6 [132].

*5.4. Involvement of the Lateral Habenula (LHb)*

According to neuroimaging and post-mortem research, in mental disorders such as major depression and schizophrenia, neuroinflammatory responses in the brain are not confined to a small area [133–135]. The results of this type of research in schizophrenia are confusing [135,136], and we also omit them for the reasons previously mentioned. The activation of microglia in the corticoid amygdala, the hippocampus, the anterior cingulate cortex and the insula described in MDD [133] can, with some imagination, be related to the functioning of the lateral habenula: these structures are directly (aCg–sCg–aIns continuum) or indirectly (amygdalo-hippocampal and septo-hippocampal systems) connected to the LHb [23,137] and play a clear role in depression [14]. Nevertheless, I consider it less likely that these changes are primary (Table 2). There is nothing to suggest specific changes in the function of these structures that by themselves should result in a depressive state. They are simultaneously involved in legion of behavioral regulatory processes, and why exactly would they result in depressive mood change by influencing the LHb? There is actually something to be said for the reverse, especially in relation to the many indications for the involvement of the LHb in the antidepressant effects of ketamine [23]. The lateral habenula regulates the activity of ascending dopaminergic pathways that run broadly from the midbrain to the aforementioned structures in the forebrain [14,22]. Dopamine D2 receptor activation exerts an important inhibitory influence on the pro-inflammatory activation of neuroglia [138]. By reducing the activity of ascending dopaminergic pathways, the LHb could disinhibit neuroinflammatory responses in the aforementioned cortical forebrain regions. Microglia express many G-protein-coupled receptors [139], but it is noteworthy that research on neuroinflammatory processes in the mental disorder schizophrenia, which is routinely treated with dopamine D2 receptor antagonists, shows so many varying results; this could be related to the anti-inflammatory effect of the described dopamine D2 antagonism.

However, what about the chronic (social) stress-related neuroinflammation seen in various animal models of depression? I would postulate that, at least in these animal models, the peripheral immune response is primary. This type of affliction is accompanied by the elevation of sympathetic activity, which, in turn, results in the activation of several types of peripheral immune cells [140–142]. In all kinds of depression, the elevation of peripheral TNF-α and IL-6 is consistently found in humans as we have seen above. TNF-α is a product of activated microglia but also brings them into this pro-inflammatory M1 state; this can trigger the whole neuroinflammatory process intracerebrally [83–85]. As an aside,

it should be noted that IFN-γ is a strong inducer of TNF-α expression by microglia [143], but for this cytokine, the results are more equivocal. It is, therefore, possible that there is another intermediary involved. Especially in the compact and highly complex regulatory system formed by the LHb, immune activation can have profound consequences for the activity of various ascending midbrain pathways. In particular, the disruption of the balance between glutamate and GABA, co-transmitters in habenula-projecting pallidum (GPh) neurons [144–146], among others, can lead to the modification of the effect of the input. In addition, neurotransmission in glutamatergic synapses can be modulated in both the short and long term. These effects are not only carried out by the (products of) microglia themselves, but also result from immunologically induced modulation by astrocytes [59,147].

**Table 2.** Putative involvement of lateral habenula in the mechanism of depression.

| Primary | Secondary | Likelihood |
|---|---|---|
| Cortical activation | - aCg–sCg–aIns gives direct input to LHb<br>- AHS gives indirect input to LHb<br>- SHS gives indirect input to LHb | Unlikely |
| LHb activation | - Microglial M1 disinhibition in striatum and cortex by ascending dopaminergic pathways | possible |
| Peripheral cytokines (TNFα, IL-6) | - Via choroid plexus to LHb microglia M1<br>- Via BBB to LHb microglia M1<br>- Via neurons and astrocytes to LHb microglia | likely |

Abbreviations: aCg—anterior cingulate cortex; AHS—amygdalo-hippocampal system; aIns—anterior insular cortex; BBB—blood–brain barrier; LHb—lateral habenula; sCg—subgenual cingulate cortex; SHS—septo-hippocampal system.

In metabolic syndrome, the interaction with and between hormones and immune factors should also be considered. The hormones involved in this case are mainly leptin, insulin, adiponectin and ghrelin, as well as immune factors which in this case are apolipoproteins in addition to cytokines [130,132,148]. Microglia carry receptors for leptin and ghrelin, the former activating these cells pro-inflammatory and the latter counteracting activation and having a neuroprotective effect [149]. Adiponectin has mutually different effects on microglia via the activation of adiponectin receptors (AdipoR1 and AdipoR2) [150]. AdipoR1 agonists inhibit the pro-inflammatory M1 state [151,152], whereas AdipoR2 activation mediates polarization to the anti-inflammatory and neuroprotective M2 state [150]. Microglia function is still most commonly associated with obesity and thus with the development of neurodegenerative diseases [89]. Under homeostatic conditions, leptin inhibits food intake by binding to receptors on specialized subsets of neurons in several hypothalamic and brainstem nuclei [149]. However, in obesity, resistance to the effects of leptin occurs [153], and this has been attributed by some to the aberrant activation of neuroglia that damages neurons [89]. Interestingly, type 2 diabetes mellitus is often associated with depression [154,155]. It is not a strange thought that high leptin levels in obesity also lead to pro-inflammatory activation in the LHb which could lead to the neuroinflammatory changes that are appropriate for depression. Moreover, the LHb also regulates appetitive reward-related processes and these may also play a role in causing obesity.

*5.5. Involvement of the Medial Habenula (MHb)*

As described above, there are quite a few differences between the MHb and the LHb. The MHb is much more compactly built, has a different, non-stellate type of astrocytes, receives input mainly from a different, restricted area (posterior septum), and gives output to a different region, i.e., mainly to the interpeduncular nucleus [33]. It is possible that the MHb is also phylogenetically older and more original than the LHb; this is a suggestion

made because the cholinergic connections of the MHb were already dominantly present in ancestors of the first vertebrates [2]. What is relevant in the context of these descriptions is that some MHb neurons express the pro-inflammatory IL-18 [67,68], and that, within the MHb, ATP for rapid neurotransmission is released from fibers derived from the posterior septum [49]. This suggests that the activation of afferent fibers may result directly in the initiation of inflammatory changes, at least those neuroplastic effects that may also be associated with the activation of immune cells. I have not found more details on this anywhere, but the MHb has been studied much less extensively than the LHb [156]. Nicotine receptors play an important role within the medial DDCS [157] and much research has been conducted on the effect that chronic smoking has on peripheral levels of cytokines [115]. This does not yet help much because chronic smoking involves many inflammatory responses. The MHb is particularly enriched in the nicotinic acetylcholine receptor (nAChR) subunits α5, α3 and β4, which are encoded by the *CHRNA5-A3-B4* gene cluster [157]. It is possible that the administration of nicotine to smoking-naive patients belonging to specific genotypes of this genetic cluster could yield something [23]. Finally, it should be noted that in the dove, mast cells may play a role in the MHb: they are found only in the MHb and their number changes according to the quantity of sex steroids in the body [158]. All in all, a lot of work still needs to be completed in this field before more clarity can be achieved.

## 6. Personal Commentary

When the entire field is considered, there seems to be quite a case for postulating that neuroinflammation-related changes have a role to play in the functioning of the DDCS as a mechanism of some forms of depression. This includes, in particular, depression due to chronic (social) stress in animal models. However, the research has many limitations; for example, attention to the habenula in this type of study may be considered niche. The complexity of the habenuloid complex and its small size also make it difficult to investigate the role of the DDCS, while the MHb is even many times smaller than the LHb. The seniority of this regulatory system of primary emotional processes (appetitive, defensive and reproductive emotional responses) argues very much for the involvement of the DDCS. Assuming that something such as "mood" in its evolutionary origin was a parameter for perceiving the situation in an individual's biotope, it is plausible that early neuronal structures also play a role in the regulation of mood in humans.

The role of the MHb seems to be a different one from that of the LHb. This inference is at least supported by the differences in construction, but also the immune response in the MHb is probably initiated in a different way to that in the LHb. However, specific research on this is even scarcer than that on the LHb.

Although this too is speculative, in the immunological modulation of connectivity via the LHb, the peripheral immune response appears to be primary. How the role of astrocytes compares with that of microglia is unclear, but there is much to be said for eventual neuromodulation by (products of) astrocytes. We are also in the dark about what the effects of the infiltration of various immune cells into the LHb are and to what extent they contribute.

## 7. Conclusions and Future Prospects

The complexity of the wiring of the DDCS in the habenular connection center makes this structure a good candidate for mediating the behavioral components of inflammatory processes. Conversely, at least some aspects of some mental disorders may be mediated by neuroinflammation that influences the habenula. The habenula receives information from parts of the forebrain responsible for generating and modulating emotional response, such as the amygdaloid and hippocampal complexes, and the continuum of limbic cortices, including aCg, sCg and aIns. To illustrate, the sCg is hyperactive in depression and is pacified when mood improves [159,160]. It is also a good target for deep brain stimulation in therapy-resistant depression [161,162]. The mentioned aCg–sCg–aIns continuum is also part of the ventral extrapyramidal re-entry circuit via the nucleus accumbens, ventral

pallidum and the thalamus: the circuits regulating pleasure and happiness. The activity of these circuits is determined by ascending monoaminergic pathways that, in turn, are controlled from the lateral habenula. It is extremely interesting to measure the effects of the glutamatergic connections between the aCg–sCg–aIns continuum and the lateral habenula, and also to determine the influence of the local application of TNFα and IL-6 on them. This could very well elucidate the mechanism by which symptoms of depression have a neuroinflammatory basis.

Another structure that I believe should be investigated further is the projection of the GPh from Figure 1 onto the lateral habenula. The GPh is well defined in representatives of the very first vertebrates and plays a role there in selecting behaviors to be continued and other behaviors to be aborted [47]. Exactly which structures are homologous to this GPh in rodents and primates has not yet been fully explored, but it may be suspected that cell bodies of these pallidohabenular neurons can be found in both the primary pallidum (bed nucleus of the stria terminalis), the secondary pallidum (ventral pallidum), and the tertiary pallidum (Globus pallidus). The input to the GPh comes from the striosomal compartment of the striatum [163,164]. Input to these pallidohabenular neurons could also be provided by the sCg [165]. Interestingly, the activity of some excitatory pallidohabenular neurons is restricted by the co-transmission of GABA, which decreases upon cocaine withdrawal due to the reduction of the presynaptic vesicular GABA transporter [146]. This results in a shift in the output of the LHb. It should be investigated whether cytokines can cause a similar shift in activity.

The organization and structure of the LHb is distinct from that of the MHb. For the LHb, some support is found in animal models in terms of depression using chronic (social) stress in addition to the interpretation of the measurement of peripheral cytokine levels, but for the MHb, all data are as yet lacking. The MHb receives input from the posterior septum, but it is not yet well understood where the disynaptic input comes from. Input to the MHb is delivered by cholinergic fibers that activate nicotinic receptors with subunits α5, α3 and β4, which are encoded by the CHRNA5-A3-B4 gene cluster [157]. By genotyping nonsmoking subjects for this cluster and selecting individuals who are carriers of functional variants, interesting clinical pharmacological experiments can be performed [23] to gain insight into the specific role of the MHb in the symptoms of mood disorders. Some septohabenular neurons use ATP as a fast transmitter [49]. In addition, interleukin-18 is secreted by MHb neurons [67,68]. The relationship between and function of these cholinergic, purinergic and IL-18-producing neurons needs to be explored, which may include interaction with the immune system.

I would like to end this paper with a call to actively investigate the significance of the habenuloid complex for the development of neuropsychiatric diseases, giving high priority to its interaction with the immune system.

**Funding:** This research received no external funding.

**Institutional Review Board Statement:** Not applicable.

**Informed Consent Statement:** Not applicable.

**Data Availability Statement:** Not applicable.

**Acknowledgments:** This article was able to emerge thanks to years of cooperation and many fruitful discussions with Svetlana S. Ivanova who is very dear to me.

**Conflicts of Interest:** The author declares no conflict of interest.

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
