# Peer review of "Role of Neuroglia in the Habenular Connection Hub of the Dorsal Diencephalic Conduction System"

_2571-6980, doi:10.3390/neuroglia4010004_

Round 1

Reviewer 1 Report

The article is well written and the author reviewed about the dorsal diencephalic connection system in conduction system.

In section 3, paragraph 2, the elaboration about the anatomy of DDCS is not clear and the flow of the article is missing. The author needs to rewrite the paragraph for clear understanding of the reader.

Further in section 3, the author explains about the relationship between microglia, astroglia and neurons in context to depression. I strongly suggest the author to tabulate the findings in all the sections. 

A detailed study of differences between the MHb and LHb is encouraged.

A final tabulation of the findings and study of the review article would be great.

Reviewer 2 Report

This manuscript provides a comprehensive review of the factors affecting habenula functions in psychiatric diseases and mental disorders. Despite the lack of evidence, the authors are trying to oversee a wide range of information about the connectivity of the DDCS and the regulation of neural activity by glia via cytokinesis, focusing on the habenula. 

The manuscript is well-written but personal excuses are not necessary (Line 57-62, 475-479).

Comment

Sometimes, it isn't easy to find some consideration of specific evidence with detail, so much just listing reference papers. Even if it is a personal reflection, it should be logically expressed with evidence from past literature.

Reviewer 3 Report

Dear Editor,

I think this manuscript is excellent and a valuable addition to the literature that focuses on complex neuropathways such as the "dorsal diencephalic conduction system” (DDCS). The DDC is not yet well understood but recent studies have shown that it may have a role in psychiatric disorders including sleep homeostasis, stress reactivity, anxiety, analgesia and schizophrenia. These types of review articles or experiments are the key to understanding the mechanisms of habenular function to design effective pharmacological and interventional treatments for depression.

In the introduction section the authors start with a short note on the role of DDCS in the development of mental illness with substantial studies, which I think is good because I immediately know what I'm about to read. The authors also mentioned the purpose of review articles with updated information on structural and functional biomarkers of DDCS. Overall, the review manuscript is interesting and insightful, and well-written; this review has the potential to be accepted. I have only a few comments and suggestions for the authors of the manuscript, which is given below:

1.     Line number "63" Title of point number 2: The Theory very briefly- Change the title or rewrite the title again.

2.     Line number 77 (Point number 2) “The activity of the basal...” not a single reference until the very end, the author cites previous and recent literature with similar findings and compares their findings with his hypothesis.

3.     Add Flow diagram in 4.4. Involvement of the LHb section that illustrates the differences between the molecular or cellular mechanisms of LHB during mental disorder and normal stage.

4.     Conclusion: This title could be changed to Conclusion and Future Prospects. In this section the author may refer to future possible investigations of LHB activity at the molecular, cellular and circuit levels using in vivo, in vitro, animal and human models.

5.     Also author can describe in the above section that continuous efforts are needed in future to understand how LHb becomes hyperactive in depression. Specifically, what kind of plasticity mechanisms lead to prolonged alteration of LHb neural activity in depression. What bio-molecule substance can be targeted for selective suppression of this hyperactivity?

Thanking you
